# Mono-InternVL: Pushing the Boundaries of Monolithic Multimodal Large Language Models with Endogenous Visual Pre-training

## Abstract

The rapid advancement of Large Language Models (LLMs) has led to an influx of efforts to extend their capabilities to multimodal tasks. Among them, growing attention have been focused on monolithic Multimodal Large Language Models (MLLMs) that integrate visual encoding and language decoding into a single LLM. Despite the structural simplicity and deployment-friendliness, training a monolithic MLLM with promising performance still remains challenging. In particular, the popular approaches adopt continuous pre-training to extend a pre-trained LLM to a monolithic MLLM, which suffers from catastrophic forgetting and leads to performance degeneration. In this paper, we aim to overcome this limitation from the perspective of delta tuning. Specifically, our core idea is to embed visual parameters into a pre-trained LLM, thereby incrementally learning visual knowledge from massive data via delta tuning, *i.e.,* freezing the LLM when optimizing the visual parameters. Based on this principle, we present Mono-InternVL, a novel monolithic MLLM that seamlessly integrates a set of visual experts via a multimodal mixture-of-experts structure. Moreover, we propose an innovative pre-training strategy to maximize the visual capability of Mono-InternVL, namely Endogenous Visual Pre-training (EViP). In particular, EViP is designed as a progressive learning process for visual experts, which aims to fully exploit the visual knowledge from noisy data to high-quality data. To validate our approach, we conduct extensive experiments on 16 benchmarks. Experimental results not only validate the superior performance of Mono-InternVL compared to the state-of-the-art MLLM on 6 multimodal benchmarks, *e.g.,* +113 points over InternVL-1.5 on OCRBench, but also confirm its better deployment efficiency, with first token latency reduced by up to 67%. Our code and models will be released.

## 1 Introduction

In recent years, the rapid development of Large Language Models (LLMs) (OpenAI, 2023; Bai et al., 2023a; Cai et al., 2024) has spurred increasing efforts to extend their multimodal capabilities (Chen et al., 2023; Liu et al., 2023e). As shown in Fig. 1 (a), most existing Multimodal Large Language Models (MLLMs) adopt a modular architecture, where visual encoding and language decoding are processed separately. This is typically achieved by combining a pre-trained visual encoder like CLIP-ViT (Radford et al., 2021) with an LLM (Liu et al., 2023e; Chen et al., 2024c; Li et al., 2023a). Recent research has also started exploring monolithic MLLMs (Bavishi et al., 2023; Diao et al., 2024; Chen et al., 2024b), as shown in Fig. 1 (b), which integrate visual perception and multimodal understanding directly into a single LLM. Due to their simplicity and unity, monolithic MLLMs can be more easily deployed using existing LLM inference libraries (LMDeployContributors, 2023) and show superior efficiency than modular MLLMs (Diao et al., 2024; Chen et al., 2024b).

Despite recent progress, training a monolithic MLLM with promising performance still remains challenging. In particular, monolithic MLLMs struggle to replicate the successful usage of pre-trained visual encoders in modular MLLMs for visual perception. Therefore, researchers often rely on additional pre-training to compensate for the shortcomings in visual perception in monolithic MLLMs (ChameleonTeam, 2024; Diao et al., 2024). A straightforward approach is the *native pre-training* (ChameleonTeam, 2024), which pre-trains a monolithic MLLM from scratch using a

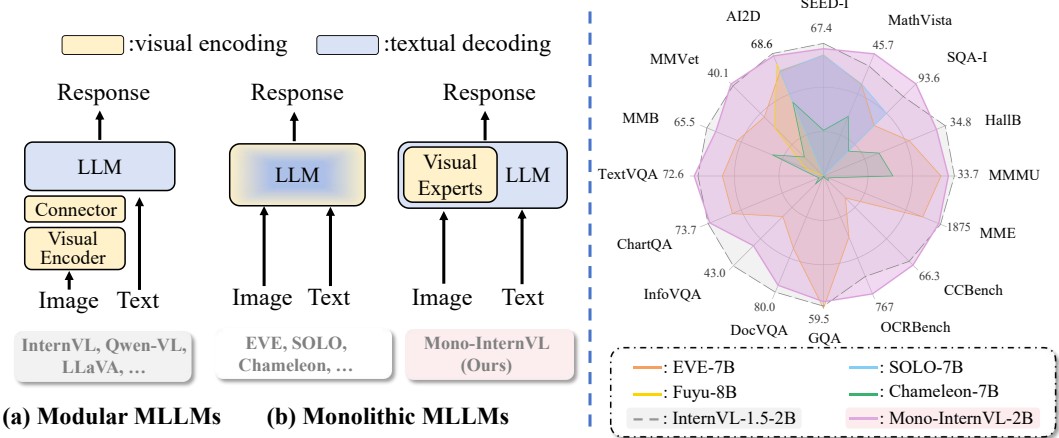

Figure 1: **Comparison of Mono-InternVL and existing MLLMs.** Compared with modular MLLMs, Mono-InternVL embeds visual experts into the pre-trained LLM and integrates visual encoding and language decoding into a single LLM. Through endogenous visual pre-training, Mono-InternVL significantly pushes the performance boundaries of monolithic MLLMs.

mixture of text-only and multimodal data. However, it requires prohibitively high training costs and often suffers from challenges of unstable optimization (ChameleonTeam, 2024). Another common solution is the *continuous pre-training* (Diao et al., 2024), which extends the pre-trained LLM to multimodality via additional visual pre-training. Benefiting from the knowledge in the pre-trained LLM, the cost of continuous pre-training becomes much more affordable. Nevertheless, due to the catastrophic forgetting issue (Zhai et al., 2023), the pre-trained language knowledge is inevitably undermined during continuous pre-training, thereby weakening the multimodal capabilities.

In this paper, we aim to address the forgetting issue of continuous pre-training through *delta tuning* (Ding et al., 2022). In particular, we argue that such issue arises from the shared architecture for joint vision and language modeling, where optimizations for vision can negatively impact language capabilities. Therefore, it is a natural thought to introduce an independent visual parameter set into the pre-trained LLM, which allows formulating visual pre-training with partial parameter tuning. This can help retain the language knowledge by freezing the entire LLM during continuous pre-training, while improving visual learning. This principle is also aligned with previous endeavors in modular MLLMs, *e.g.,* QwenVL (Bai et al., 2023b) and InternVL-1.5 (Chen et al., 2024c), where the visual parameters are placed outside the LLM.

Based on the above principle, we propose a novel monolithic MLLM, namely Mono-InternVL. As shown in Fig. 2, the visual parameters in Mono-InternVL are designed as a set of expert networks that are seamlessly integrated into the LLM via the mixture-of-experts mechanism. To better capture the visual knowledge, these experts are initialized from the Feed-Forward Networks (FFNs) in the pre-trained LLM. Based on this architecture, we present an innovative visual pre-training method called *Endogenous Visual Pre-training* (EViP). Specifically, EViP is formulated as a progressive learning process of three stages: 1) concept learning to grasp basic visual concepts, 2) semantic learning to capture high-level semantics, *e.g.,* world knowledge, and 3) alignment learning to align knowledge with downstream tasks. Benefiting from the architecture and the pre-training strategy, the visual scalability of Mono-InternVL is fully unleashed, where the downstream performance consistently improves as the scale of the pre-training data increases. After visual pre-training, Mono-InternVL accommodates complex multimodal tasks via supervised instruction tuning.

To validate our method, we develop Mono-InternVL using the pre-trained LLM InternLM2-1.8B (Cai et al., 2024), and conduct extensive experiments on 16 multimodal benchmarks. Experimental results not only demonstrate the significant performance improvements of Mono-InternVL against previous monolithic MLLMs, but also validate its superior efficiency compared to existing modular MLLMs. For instance, Mono-InternVL with 1.8 billion activated parameters can outperform existing monolithic MLLMs with 7 billion parameters by a significant margin, *e.g.,* +15.5% over EVE on average. Compared to the state-of-the-art modular MLLM InternVL-1.5 (Chen et al.,

2024c), Mono-InternVL shows superior performance on 6 multimodal benchmarks while reducing first token latency by 67%. In conclusion, our contributions can be summarized in threefold:

- We present Mono-InternVL, a novel monolithic MLLM that seamlessly integrates a set of visual experts via a multimodal mixture-of-experts architecture. This architecture can effectively extend the pre-trained LLM to a monolithic MLLM while retaining the pre-trained knowledge.

- We propose a novel visual pre-training approach for Mono-InternVL called endogenous visual pre-training (EViP). EViP adopts a progressive learning strategy to encourage visual experts of Mono-InternVL to continuously grasp visual knowledge from noisy data to high-quality data.

- Mono-InternVL demonstrates for the first time that the leading performance of MLLM no longer depends on the pre-trained visual encoder, thereby opening new avenues for designing future MLLMs. In particular, Mono-InternVL achieves the state-of-the-art (SoTA) results compared to existing MLLMs on 6 multimodal benchmarks.

## 2 RELATED WORK

**Modular multimodal large language models.**  Recent progress in large language models (LLMs) has catalyzed the integration of vision and language modalities, giving rise to multimodal large language models (MLLMs). Both commercial entities, such as GPT-4V (Yang et al., 2023) and Gemini series (Team et al., 2023), and other open-source initiatives, *e.g.* BLIP series (Li et al., 2022; 2023a; Dai et al., 2023), LLaVA series (Liu et al., 2023e;d; 2024), InternVL (Chen et al., 2023; 2024c), have pursued this integration, often linking LLMs (Touvron et al., 2023a;b; Team, 2023; Cai et al., 2024) with large vision models (LVMs) (Radford et al., 2021; Dosovitskiy et al., 2021; Dehghani et al., 2023; Chen et al., 2023) via intermediate layers. Specifically, there are lightweight MLLMs (*i.e.,* $\leq$ 4B parameters), such as MobileVLM-V2 (Chu et al., 2024), Mini-Gemini (Li et al., 2024), MM1 (McKinzie et al., 2024), DeepSeek-VL (Lu et al., 2024), PaliGemma (Beyer et al., 2024) and MiniCPM-V (Yao et al., 2024). However, such encoder-based vision-language models (modular MLLMs) encounter challenges like limitations in input processing due to pre-trained visual encoders, deployment inefficiencies, and complexities in balancing the capacities of LLMs and LVMs, as also pointed out by (Diao et al., 2024).

**Monolithic multimodal large language models.**  The issues related to modular MLLMs have steered research toward encoder-free architectures, also known as monolithic MLLMs, which can be summarized into two categories. The first category obtains continuous visual tokens through a lightweight structure before feeding into MLLMs. For instance, Fuyu-8B (Bavishi et al., 2023) processes images directly through a simple linear projection, adeptly handling high-resolution images without using a dedicated visual encoder. EVE-7B (Diao et al., 2024) prioritizes vision-language pre-alignment from an LLM-centric perspective and enhances image recognition through visual representation supervision. SOLO (Chen et al., 2024b) introduces an open-source training recipe for developing monolithic MLLMs. In contrast, the second category introduces VQ tokenizer-based models to generate discrete visual tokens to support image generation tasks, with representative works such as Chameleon (ChameleonTeam, 2024), Show-o (Xie et al., 2024), Transfusion (Zhou et al., 2024), and Emu3 (Wang et al., 2024b).

**Multimodal mixture-of-experts.**  VLMo (Bao et al., 2022) and BEiT-3 (Wang et al., 2022) employ a pool of modality experts to replace the feed-forward network in standard Transformer to capture modality-specific information by switching to different modality experts, and use the shared self-attention across modalities to align visual and linguistic information. VL-MoE (Shen et al., 2023) introduces mixture-of-experts (MoE) (Yuksel et al., 2012) based on the above works for better training and deploying. MoMa (Lin et al., 2024) also uses multimodal mixture-of-experts for pre-training of MLLMs (ChameleonTeam, 2024) and cooperates with sparse components, *e.g.* MoE and mixture-of-depths (MoD) (Raposo et al., 2024) to improve the efficiency of pre-training from scratch with trillions of mixed-modal tokens. Inspired by the above literature, we introduce multimodal mixture-of-experts (*i.e.,* a visual expert and a language expert) for pre-training monolithic MLLMs and use a novel progressive learning strategy, named endogenous visual pre-training (EViP), to address the unique challenges faced by monolithic MLLMs.

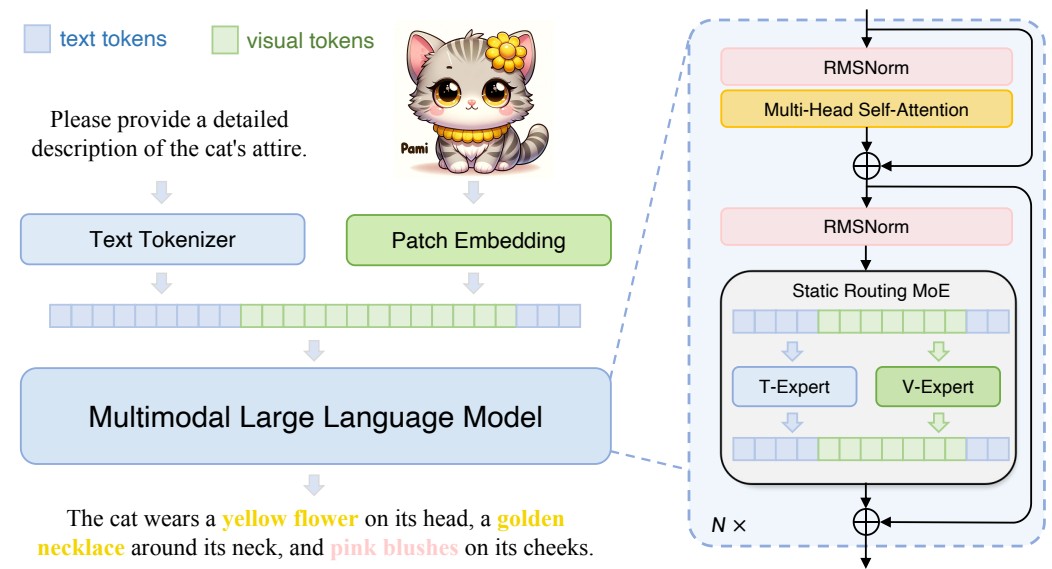

Figure 2: **Monolithic architecture of Mono-InternVL.** Mono-InternVL is designed as a multimodal MoE structure, where visual and textual tokens are processed by the corresponding experts. Such a design greatly facilitates the visual pre-training while retaining the model efficiency.

## 3 MONO-INTERNVL

### 3.1 THE MONOLITHIC ARCHITECTURE

As shown in Fig. 2, we first outline the architecture of Mono-InternVL, which consists of tokenizers and a multimodal mixture-of-experts structure.

**Visual and textual embeddings.** Compared to modular MLLMs, Mono-InternVL directly patchifies images to input visual sequences using a lightweight module. Specifically, given the input image $I \in \mathbb{R}^{H \times W \times 3}$, the input visual embedding $x_v \in \mathbb{R}^{(h \times w) \times d}$ is obtained by

$$x_v = \text{MLP}(\text{PatchEmbed}(I) + \text{PE}). \tag{1}$$

Here, PatchEmbed$(\cdot)$ denotes a patch embedding layer with a stride of 28, meaning each visual token represents a $28 \times 28$ image patch. PE $\in \mathbb{R}^{(h \times w) \times d}$ is the learnable positional embedding as similar to InternVL-1.5 (Chen et al., 2024c). Besides, we also add an additional thumbnail to provide global visual information. After that, an MLP layer, *i.e.,* MLP$(\cdot)$, is used to project visual patches into the $d$-dimensional embedding space of the LLM. This simple visual tokenizer allows Mono-InternVL to process arbitrary-resolution images with up to 8 millions of pixels, *i.e.,* $10,240$ image patches, which can cover most high-resolution scenarios.

In Mono-InternVL, the textual tokenizer remains the same as the original one in the LLM. In particular, given the input text $T \in \mathbb{Z}^n$, we obtain textual embedding $x_t \in \mathbb{R}^{n \times d}$ by

$$x_t = \text{Tokenizer}(T). \tag{2}$$

Afterwards, the multimodal embedding is constructed as the concatenation of visual and textual embeddings, denoted as $x_m \in \mathbb{R}^{n' \times d}$.

**Multimodal mixture-of-experts structure.** The key principle of Mono-InternVL is to embed visual experts into a pre-trained LLM. In this case, Mono-InternVL can not only facilitate the visual pre-training with the pre-trained LLM knowledge, but also significantly mitigates the catastrophic forgetting issue during pre-training. Specifically, given the multimodal input $x_m \in \mathbb{R}^{n' \times d}$, a decoder-only LLM with a set of visual experts is used to generate the textual tokens step by step, which can be formulated by

$$p_s = \mathcal{F}_{\text{llm}}(y_s | x_m, y_{0:s-1}; \theta, \theta_v). \tag{3}$$

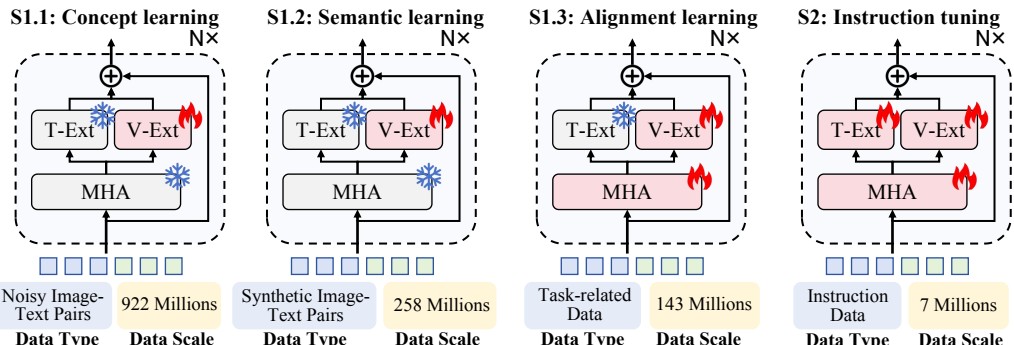

Figure 3: **The training recipe of Mono-InternVL.** In the first stage, Mono-Internvl is progressively pre-trained on massive data via three sub-stages (S1.1, S1.2, S1.3), where most parameters of LLM are frozen to preserve the pre-trained knowledge. In the second stage (S2), the entire model is optimized to accommodate various instructions.

Here, $y \in \mathbb{R}^S$ and $S$ denote the word length and its length, respectively. $p_s \in \mathbb{R}^m$ is the next-token probability and $m$ is the size of word vocabulary. $\mathcal{F}_{\text{llm}}$ and $\theta$ denote the LLM and its pre-trained parameters, respectively. $\theta_v$ is the parameters of patch embedding layer and visual experts.

As shown in Fig. 2, $\mathcal{F}_{\text{llm}}$ is designed as a multimodal mixture-of-experts structure. In particular, we adopt the static routing strategy that assigns visual and textual experts to the corresponding tokens. Therefore, the $l$-th LLM layer can be defined by

$$
\begin{aligned}
x_m^{l'} &= x_m^{l-1} + \text{MHA}(\text{RMSNorm}(x_m^{l-1})), \\
x_m^{l} &= x_m^{l'} + \text{MMoE}(\text{RMSNorm}(x_m^{l'})).
\end{aligned}
\tag{4}
$$

Here, MHA($\cdot$) and RMSNorm($\cdot$) denote the multi-head attention (Vaswani et al., 2017) and the layer normalization (Zhang & Sennrich, 2019), respectively. MMoE($\cdot$) is the proposed multimodal mixture-of-experts, formulated by

$$
\text{MMoE}(x) = \begin{cases} \text{FFN}_v(x) & \text{if } x \in x_v, \\ \text{FFN}_t(x) & \text{if } x \in x_t. \end{cases}
\tag{5}
$$

Here, $x \in \mathbb{R}^d$ is the element of $x_m$. $\text{FFN}_v$ and $\text{FFN}_t$ denote the visual and textual experts, respectively. In practice, $\text{FFN}_v$ is initialized from the $\text{FFN}_t$ to leverage the pre-trained knowledge.

As defined in Eq. 4 and 5, the MMoE structure has two distinct advantages over the existing monolithic MLLM. Firstly, the visual learning of Mono-InternVL can greatly benefit from the pre-trained language knowledge, while the language ability can still be preserved by freezing $\text{FFN}_t$. Secondly, the MMoE structure significantly enhances the model's capacity for vision-and-language modeling, while the additional inference cost is almost negligible due to the MoE mechanism.

## 3.2 ENDOGENOUS VISUAL PRE-TRAINING

Endogenous Visual Pre-training (EViP) aims to maximize the benefits of Mono-InternVL from visual experts through pre-training on massive noisy and synthetic data. Unlike existing methods (Diao et al., 2024; ChameleonTeam, 2024), we formulate EViP from the perspective of delta tuning (Ding et al., 2022), in which most of the LLM parameters are frozen to preserve its pre-trained knowledge. Therefore, the objective of EViP can be defined by

$$
\arg \min_{\Delta \theta} \mathcal{L}(\mathcal{F}_{\text{llm}}(x_m; \theta, \theta_v), \hat{y}),
\tag{6}
$$

where $\mathcal{L}(\cdot)$ and $\hat{y}$ denote the auto-regressive loss and the ground-truth, respectively. As shown in Fig. 3, $\Delta \theta$ denotes parameters of patch embedding and visual experts in the concept and semantic learning, *i.e.,* $\theta_v$, while in the alignment learning stage $\Delta \theta$ also includes the parameters of multi-head attentions. Based on Eq. 6, EViP is designed as a progressive learning process. As shown in Fig. 3 and Tab. 1, EViP consists of three sub-stages, namely concept learning (S1.1), semantic

Table 1: **Summary of datasets used in the endogenous visual pre-training.** In S1.2, caption for each image is synthetically produced by the pre-trained InternVL-8B (Chen et al., 2024c).

| Stage | #Samples | Datasets |
|---|---|---|
| S1.1 | 922M | Laion-EN (en) (Schuhmann et al., 2022a), COYO (en) (Byeon et al., 2022) |
| S1.2 | 258M | Laion-EN (en) (Schuhmann et al., 2022a), COYO (en) (Byeon et al., 2022), SAM (en) (Kirillov et al., 2023) |
| S1.3 | 143M | **Captioning:** Laion-EN (en) (Schuhmann et al., 2022a), Laion-ZH (zh) (Schuhmann et al., 2022a), COYO (zh) (Byeon et al., 2022), GRIT (zh) (Peng et al., 2023), COCO (en) (Chen et al., 2015), TextCaps (en) (Sidorov et al., 2020)
**Detection:** Objects365 (en&zh) (Shao et al., 2019), GRIT (en&zh) (Peng et al., 2023), All-Seeing (en&zh) (Wang et al., 2024a)
**OCR (large):** Wukong-OCR (zh) (Gu et al., 2022), LaionCOCO-OCR (en) (Schuhmann et al., 2022b), Common Crawl PDF (en&zh)
**OCR (small):** MMC-Inst (en) (Liu et al., 2023c), LSVT (zh) (Sun et al., 2019), ST-VQA (en) (Biten et al., 2019), RCTW-17 (zh) (Shi et al., 2017), ReCTs (zh) (Zhang et al., 2019), ArT (en&zh) (Chng et al., 2019), SynthDoG (en&zh) (Kim et al., 2022), COCO-Text (en) (Veit et al., 2016), ChartQA (en) (Masry et al., 2022), CTW (zh) (Yuan et al., 2019), DocVQA (en) (Clark & Gardner, 2018), TextOCR (en) (Singh et al., 2021), PlotQA (en) (Methani et al., 2020), InfoVQA (en) (Mathew et al., 2022) |

learning (S1.2) and alignment learning (S1.3). For different sub-stages, we use carefully partitioned data to achieve the coarse-to-fine visual learning.

**Concept learning.** Concept learning aims to encourage the model to learn fundamental visual concepts, such as object categories or basic shapes. Therefore, we first pre-train Mono-InternVL with about 922 million noisy samples, which are sampled from Laion-2b (Schuhmann et al., 2022a) and Coyo-700M (Byeon et al., 2022). In this sub-stage, Mono-InternVL employs a simple prompt to perform generative learning, *i.e.,* "provide a one-sentence caption for the image". Meanwhile, we constrain the maximum number of image patches of the visual tokenizer to 1,280 for training efficiency. To ensure that the foundational language capabilities are preserved while enabling visual specialization, the entire LLM is kept frozen during concept learning, and only the patch embedding and visual experts are optimized.

**Semantic learning.** After concept learning, Mono-InternVL is able to understand basic concepts in the image, but organizing this information to produce reasonable descriptions remains challenging. To achieve a higher-level visual understanding, we utilize the pre-trained InternVL-8B (Chen et al., 2024c) to produce short captions for 258 million images. Compared to the original noisy captions, synthetic captions typically depict complex visual knowledge, such as relationship and world knowledge, *etc.*, while containing less noisy information unrelated to the image, *e.g.,* time of shooting, and the photographer. In this sub-stage, we adopt the same optimization strategy as concept learning, except that the maximum number of image patches is increased to 1,792.

**Alignment learning.** To meet the visual requirements of downstream tasks, we further perform alignment learning on Mono-InternVL. As shown in Tab. 1, our alignment data is sampled from the pre-training data of InternVL-1.5 (Chen et al., 2024c), including 143 million samples of image captioning, detection and optical character recognition (OCR). In particular, captioning data, detection data and OCR data account for about 53.9%, 5.2% and 40.9% of the total, respectively. In this sub-stage, we utilize the task-specific prompts from InternVL-1.5 for the generative learning, and increase the maximum number of image patches to 3,328. Compared to previous sub-stages, the multi-head attention layers are additionally optimized to achieve better vision-language alignment.

### 3.3 INSTRUCTION TUNING

In this stage, we follow InternVL (Chen et al., 2024c) to use around 5 million bilingual instructions for supervised learning, covering various tasks such as visual question answering, multimodal dialogue, mathematics, knowledge, *etc*. In addition to this, we further include additional instruction data for video understanding and handwritten text recognition. In this stage, the entire models are optimized and the maximum number of image patches is increased to 6,400 to accommodate high-resolution images. Details of instruction data can be found in Appendix §A.1.

Table 2: **Comparison with existing MLLMs on general MLLM benchmarks.** "#A-Param" denotes the number of activated parameters. For MME, we sum the perception and cognition scores. Average scores are computed by normalizing each metric to a range between 0 and 100.

| Model | #A-Param | MMB | MMVet | MMMU | MME | MathVista | SEED-I | OCRBench | HallB | CCB | Avg |
|---|---|---|---|---|---|---|---|---|---|---|---|
| *Modular MLLMs:* | | | | | | | | | | | |
| MobileVLM-V2-1.7B | 1.7B | 57.7 | – | – | – | – | – | – | – | – | – |
| MobileVLM-V2-3B | 3.0B | 63.2 | – | – | – | – | – | – | – | – | – |
| Mini-Gemini-2B | 3.5B | 59.8 | 31.1 | 31.7 | 1653 | 29.4 | – | – | – | – | – |
| MM1-3B-MoE-Chat | 3.5B | 70.8 | 42.2 | 38.6 | 1772 | 32.6 | 69.4 | – | – | – | – |
| DeepSeek-VL-1.3B | 2.0B | 64.6 | 34.8 | 32.2 | 1532 | 31.1 | 66.7 | 409 | 27.6 | 37.6 | 43.4 |
| PaliGemma-3B | 2.9B | 71.0 | 33.1 | 34.9 | 1686 | 28.7 | 69.6 | 614 | 32.2 | 29.6 | 46.7 |
| MiniCPM-V | 2.8B | 64.1 | 31.1 | 38.3 | 1650 | 28.9 | 65.6 | 366 | 36.2 | 41.4 | 44.6 |
| MiniCPM-V-2 | 2.8B | 69.1 | 41.0 | 38.2 | 1809 | 38.7 | 67.1 | 605 | 36.1 | 45.3 | 51.2 |
| InternVL-1.5-2B | 2.2B | 70.9 | 39.3 | 34.6 | 1902 | 41.1 | 69.8 | 654 | 37.5 | 63.5 | 54.4 |
| *Monolithic MLLMs:* | | | | | | | | | | | |
| Fuyu-8B (HD) | 8B | 10.7 | 21.4 | – | – | – | – | – | – | – | – |
| SOLO | 7B | – | – | – | 1260 | 34.4 | 64.4 | – | – | – | – |
| Chameleon-7B[1] | 7B | 31.1 | 8.3 | 25.4 | 170 | 22.3 | 30.6 | 7 | 17.1 | 3.5 | 16.1 |
| EVE-7B | 7B | 49.5 | 25.6 | 32.3 | 1483 | 25.2 | 61.3 | 327 | 21.1 | 12.4 | 34.8 |
| EVE-7B (HD) | 7B | 52.3 | 25.7 | 32.6 | 1628 | 34.2 | 64.6 | 398 | 26.4 | 16.3 | 38.9 |
| Emu3 | 8B | 58.5 | 37.2 | 31.6 | – | – | **68.2** | 687 | – | – | – |
| Mono-InternVL-2B | 1.8B | **65.5** | **40.1** | **33.7** | **1875** | **45.7** | 67.4 | **767** | **34.8** | **66.3** | **55.2** |

## 4 EXPERIMENTS

### 4.1 EVALUATION BENCHMARKS

We evaluate Mono-InternVL and existing MLLMs on 16 comprehensive multimodal benchmarks. Specifically, general MLLM benchmarks encompass MMBench-EN *test* (Liu et al., 2023f), MMVet (Yu et al., 2023b), MMMU *val* (Yue et al., 2023), MME (Fu et al., 2023), MathVista *test-mini* (Lu et al., 2023), SEED Image (Ge et al., 2024), OCRBench (Liu et al., 2023g), Hallusion-Bench (Guan et al., 2023), and CCBench *dev* (Liu et al., 2023f). Visual question answering benchmarks include TextVQA *val* (Singh et al., 2019), SQA *test* (Lu et al., 2022a), GQA *test-dev* (Hudson & Manning, 2019), DocVQA *test* (Clark & Gardner, 2018), AI2D *test* (Kembhavi et al., 2016), ChartQA *test* (Masry et al., 2022), and InfographicVQA *test* (Mathew et al., 2022). The evaluation metrics follow the existing methods (Chen et al., 2024c; Diao et al., 2024). Part of the results of Chameleon and EVE are evaluated with VLMEvalKit (Duan et al., 2024) or from the OpenCompass leaderboard (Contributors, 2023).

### 4.2 IMPLEMENTATION DETAILS

Mono-InternVL is implemented based on InternLM2-1.8B (Cai et al., 2024) with newly added visual tokenizer and visual experts. The visual experts are initialized from pre-trained MLPs in InternLM2-1.8B to leverage existing learned representations for improved visual feature extraction, which accounts for 1.2 billion parameters. We adopt a similar dynamic high-resolution strategy from InternVL-1.5 (Chen et al., 2024c) to align an optimal resolution for input image, which is then patchfied to visual tokens. The remaining configurations are kept identical to InternLM2-1.8B. The endogenous visual pre-training and instruction tuning take approximately 16 days (646*k* iterations) and 1 day (14*k* iterations) on 256 NVIDIA A100 GPUs, respectively. More detailed training configurations are given in Appendix §A.2.

### 4.3 QUANTITATIVE EXPERIMENTS

**Comparison with existing MLLMs.** In Tab. 2 and 3, we compare Mono-InternVL and existing MLLMs on 16 multimodal benchmarks. From Tab. 2, the first observation is that most modular

---

[1]Chameleon-7B frequently rejects to perform the task with a response of "I can't help you with this", thus resulting in poor performance.

Table 3: **Comparison with existing MLLMs on visual question answering benchmarks.**

| Model | #A-Param | TextVQA | SQA-I | GQA | DocVQA | AI2D | ChartQA | InfoVQA | Avg |
|---|---|---|---|---|---|---|---|---|---|
| *Modular MLLMs:* | | | | | | | | | |
| MobileVLM-V2-1.7B | 1.7B | 52.1 | 66.7 | 59.3 | — | — | — | — | — |
| MobileVLM-V2-3B | 3.0B | 57.5 | 70.0 | 66.1 | — | — | — | — | — |
| Mini-Gemini-2B | 3.5B | 56.2 | — | — | 34.2 | — | — | — | — |
| MM1-3B-MoE-Chat | 3.5B | 72.9 | 76.1 | — | — | — | — | — | — |
| DeepSeek-VL-1.3B | 2.0B | 57.8 | — | — | — | 51.5 | — | — | — |
| PaliGemma-3B | 2.9B | 68.1 | — | — | — | 68.3 | — | — | — |
| MiniCPM-V | 2.8B | 60.6 | — | — | 38.2 | 56.3 | — | — | — |
| MiniCPM-V-2 | 2.8B | 74.1 | — | — | 71.9 | 62.9 | — | — | — |
| InternVL-1.5-2B | 2.2B | 70.5 | 84.9 | 61.6 | 85.0 | 69.8 | 74.8 | 55.4 | 71.7 |
| *Monolithic MLLMs:* | | | | | | | | | |
| Fuyu-8B (HD) | 8B | — | — | — | — | 64.5 | — | — | — |
| SOLO | 7B | — | 73.3 | — | — | 61.4 | — | — | — |
| Chameleon-7B[1] | 7B | 4.8 | 47.2 | — | 1.5 | 46.0 | 2.9 | 5.0 | 17.9 |
| EVE-7B | 7B | 51.9 | 63.0 | 60.8 | 22.0 | 48.5 | 19.5 | 20.0 | 40.8 |
| EVE-7B (HD) | 7B | 56.8 | 64.9 | **62.6** | 53.0 | 61.0 | 59.1 | 25.0 | 54.6 |
| Emu3 | 8B | 64.7 | 89.2 | 60.3 | 76.3 | **70.0** | 68.6 | **43.8** | 67.6 |
| Mono-InternVL-2B | 1.8B | **72.6** | **93.6** | 59.5 | **80.0** | 68.6 | **73.7** | 43.0 | **70.1** |

Table 4: **Zero-shot pre-training performance of Mono-InternVL and existing MLLMs.** "S1.2" and "S1.3" denote pre-training stages of semantic learning and alignment learning, respectively. Images of COCO have been seen in Mono-InternVL-S1.3, so we mark its performance in gray.

| Model | #A-Param | Data | Shots | COCO Caps | Flickr30k | NoCaps | VQAv2 |
|---|---|---|---|---|---|---|---|
| Flamingo | 3B | >2.1B | 0 | 73.0 | — | — | 49.2 |
| MM1 | 3.5B | >2.3B | 0 | 73.5 | — | 55.6 | 46.2 |
| Chameleon | 34B | >1.4B | 2 | 120.2 | 74.7 | — | 66.0 |
| Mono-InternVL-S1.2 | 1.8B | 0.9B | 0 | 87.3 | 72.7 | 54.1 | — |
| Mono-InternVL-S1.3 | 1.8B | 1.1B | 0 | 135.6 | 77.3 | 116.5 | 71.1 |

MLLMs outperform existing monolithic MLLMs by significant margins. For example, the average performance of InternVL-1.5-2B (Chen et al., 2024c) on 9 MLLM benchmarks greatly exceeds the SoTA monolithic MLLM, *i.e.,* + 15.5% over EVE-7B (HD) (Diao et al., 2024). These results strongly suggest the challenges in existing monolithic MLLMs. In contrast, Mono-InternVL-2B with a slightly smaller model size can even outperform the SoTA modular MLLM, *i.e.,* + 0.8% against InternVL-1.5-2B on average. Notably, Mono-InternVL-2B demonstrates distinct advantages on MathVista and OCRBench, suggesting its seamless text recognition and reasoning capabilities. Moreover, superior bilingual ability of Mono-InternVL-2B is also validated on CCBench, which contains a large amount of questions related to Chinese culture. Compared to existing monolithic MLLMs, performance gains of Mono-InternVL are more distinct, *e.g.,* +15.4% over EVE-7B (HD) (Diao et al., 2024) on MMVet and +7.9% over Emu3 (Wang et al., 2024b) on TextVQA, while using a much smaller parameter scale. Similar advantages of Mono-InternVL can also been witnessed in Tab. 3, *e.g.,* +2.1% on TextVQA. Nevertheless, we also observe that Mono-InternVL is still inferior to InternVL-1.5 on high-resolution benchmarks, *e.g.,* -12.4% on InfoVQA. This is because specific optimizations for high-resolution encoding are not the focus of the paper, *e.g.,* positional embedding and high-resolution training data, which we plan to explore in future research. Overall, these comparisons significantly validate the architecture and training strategy of Mono-InternVL.

In Tab. 4, we further compare the pre-training performance of Mono-InternVL and existing MLLMs. From this table, we can observe that with concept and semantic learning, Mono-InternVL-S1.2 already exceeds existing modular MLLMs, *e.g.,* +13.8 CIDEr over MM1 (McKinzie et al., 2024) on COCO Captions, demonstrating that Mono-InternVL-S1.2 is effective in capturing fundamental multimodal relationships. It is worth noting that pre-training in Mono-InternVL-S1.2 only consumes

Table 5: **Ablation of different strategies for visual pre-training.** All models are pre-trained on 61 million image-text pairs from Laion-2B (Schuhmann et al., 2022a) and fine-tuned on instruction data from LLaVA-665*k*. (Liu et al., 2023d). "Full" and "Delta" denote full tuning and delta tuning, respectively. "T-Param" refers to trainable parameters.

| Model | #T-Param | Strategy | MME-P | DocVQA | InfoVQA | SQA-I | GQA | ChartQA | AI2D |
|---|---|---|---|---|---|---|---|---|---|
| InternLM2 | 1.8B | Full | 753 | 16.1 | 11.6 | 36.7 | 51.4 | 10.8 | 27.7 |
| + V-Expert | 3.0B | Full | 948 | 18.6 | 11.9 | 37.7 | 53.0 | 11.1 | 26.6 |
| + V-Expert | 1.2B | Delta | 995 | 18.9 | 14.6 | 56.5 | 53.4 | 13.5 | 42.7 |

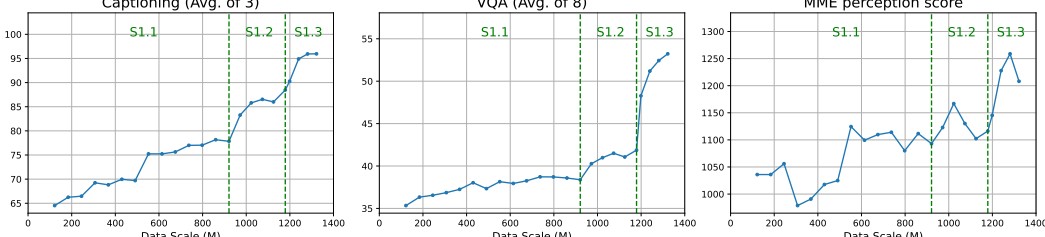

Figure 4: **Downstream performance breakdown with the increase of pre-training data size across three sub-stages: (S1.1) Concept learning; (S1.2) Semantic learning; (S1.3) Alignment learning.** For each data point, we fine-tune the corresponding pre-trained model on the instruction data of LLaVA-665*k* and obtain the downstream performance. Results of captioning and VQA are averaged from 3 and 8 tasks, respectively. See Appendix §A.3 for complete results.

about 0.9B image-text pairs, but the cost in MM1 and Flamingo is much more expensive, *e.g.,* more than 2B data. Compared to monolithic MLLMs, Mono-InternVL also demonstrates superior performance. For instance, even though Chameleon has a much larger model size, it is still inferior to Mono-InternVL-S1.3 by -2.6 CIDEr on Flickr30k (Young et al., 2014). These results further confirm the effectiveness of EViP for Mono-InternVL.

**Ablation studies.** To validate the design of Mono-InternVL, we conduct extensive ablation studies in Tab. 5 and Fig. 4. Specifically, Tab. 5 compares different strategies for visual pre-training. The first row is the common strategy used in existing monolithic MLLMs, *i.e.,* full tuning of the LLM, which yields the worst downstream performance in the table. After employing visual experts (the second row), such a full-tuning strategy becomes more effective, *e.g.,* +1.6% on GQA. These comparisons well validate that the shared architecture for joint vision and language modeling is sub-optimal in monolithic MLLMs. Besides, we also observe that the delta tuning strategy greatly benefits the visual pre-training, providing +18.8% and 16.1% gains on SQA-I and AI2D, respectively. Compared to full tuning, delta tuning can effectively preserve the knowledge of the pre-trained LLM, which is also crucial for maintaining the language understanding capabilities required for effective multimodal interactions. These comparisons clearly indicate the significance of visual experts and the delta tuning strategy.

Fig. 4 further demonstrates the relationship between downstream performance and pre-training data size. From it we can observe that performance of Mono-InternVL will gradually reach an upper bound in the concept learning. Through additional semantic learning and alignment learning, capabilities of Mono-InternVL consistently boost as the data size increases. It is important to note that that the alignment learning plays a significant role for VQA and MME, which can provide sufficient task-related knowledge, *e.g.,* OCR knowledge. These results not only demonstrate the data scalability of Mono-InternVL, but also confirm the advantages of the coarse-to-fine learning in EViP.

**Comparison of inference efficiency.** In Tab. 6, we compare the inference speed of Mono-InternVL and InternVL-1.5 using the popular deployment library LMDeploy (LMDeployContributors, 2023). From this table, we can find that due to the elimination of visual encoder, Mono-InternVL demonstrates superior efficiency under different number of input tokens. In particular, the first-token time is greatly reduced in Mono-InternVL, *e.g.,* up to -67% against InternVL-1.5. Benefiting from this, the overall throughput is correspondingly increased by around 31%. These results greatly validate the efficiency of Mono-InternVL. We note that this is only an initial attempt, and using Turbomind backend or further optimization techniques may yield better performance.

Table 6: **Inference speed comparison of Mono-InternVL and InternVL-1.5.** Models are deployed on an NVIDIA A100 using LMDeploy with Pytorch backend (LMDeployContributors, 2023), with a concurrency of 16 and the number of output tokens fixed as 120. "TTFT" and "TPS" denotes the time to first token in seconds and throughput in tokens per second, respectively.

| Model | #Image Tokens | #Text Tokens | #Total Input Tokens | TTFT | TPS |
|---|---|---|---|---|---|
| InternVL-1.5-2B | 768 | 256 | 1024 | 0.24 | 382 |
| Mono-InternVL-2B | 768 | 256 | 1024 | 0.09 (-63%) | 436 (+14%) |
| InternVL-1.5-2B | 1792 | 256 | 2048 | 0.45 | 183 |
| Mono-InternVL-2B | 1792 | 256 | 2048 | 0.15 (-67%) | 232 (+27%) |
| InternVL-1.5-2B | 3840 | 256 | 4096 | 1.93 | 52 |
| Mono-InternVL-2B | 3840 | 256 | 4096 | 0.79 (-59%) | 68 (+31%) |

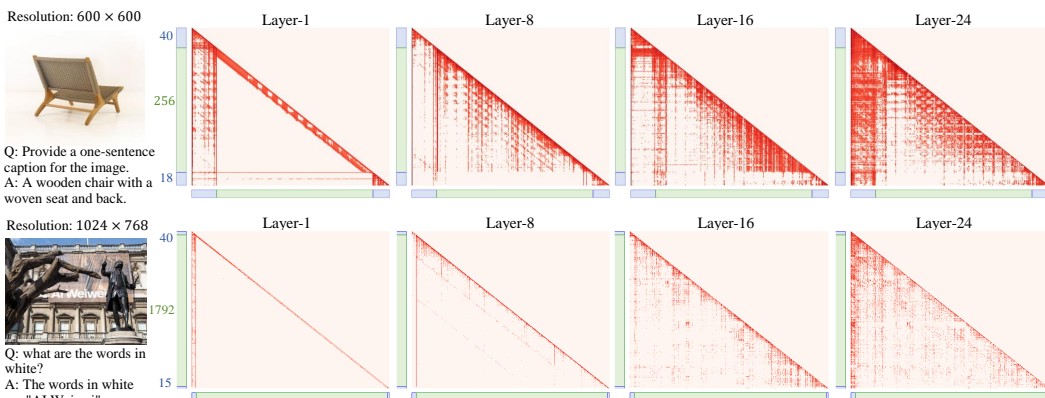

Figure 5: **Visualization of attention maps in Mono-InternVL.** The first blue segment, green segment and the second green segment in the axes represent the system prompt tokens (text), image tokens (visual) and user prompt tokens (text), respectively. The numbers on the left side of attention maps indicate the number of tokens.

## 4.4 QUALITATIVE EXPERIMENTS

To gain in-depth insights into Mono-InternVL, we visualize its attention maps of different layers in Fig. 5. From this figure, we can draw two noteworthy findings. Firstly, despite the global connectivity of the Transformer architecture, locality still exists in the visual encoding of shallow layers. As shown in Fig. 5, in the first layer, visual tokens only interact with their nearby content, yielding patterns that closely resemble those produced by convolutional neural networks (He et al., 2016). Secondly, modalities are barely interactive at shallow layers but gradually fused as the layers deepen. As illustrated in Fig. 5, the attention weights between visual and textual tokens are extremely small in the first layer and become larger in deeper layers. We believe these examples will provide useful hints for the design of monolithic MLLMs. More examples are given in Appendix §A.4.

## 5 CONCLUSION

In this paper, we propose Mono-InternVL, a monolithic multimodal large language model (MLLM) that integrates visual encoding and textual decoding into a single LLM. In Mono-InternVL, a set of visual experts is embedded into the pre-trained LLM via a mixture-of-experts mechanism. By freezing the LLM, Mono-InternVL ensures that visual capabilities are optimized without compromising the pre-trained language knowledge. Based on this structure, an innovative Endogenous Visual Pre-training (EViP) is introduced to realize the coarse-to-fine visual learning. Extensive experiments demonstrate the effectiveness and efficiency of Mono-InternVL compared to existing MLLMs. Our work greatly pushes the boundaries of monolithic MLLMs, providing new possibilities for the development of MLLMs.

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

# A APPENDIX

## A.1 MORE DATASET DETAILS

Table 7: **Summary of datasets used in instruction tuning.**

| Task | Dataset |
|---|---|
| Captioning | TextCaps (en) (Sidorov et al., 2020), ShareGPT-4o (en&zh) (Chen et al., 2024c) |
| General QA | VQAv2 (en) (Goyal et al., 2017), GQA (en) (Hudson & Manning, 2019), OKVQA (en) (Marino et al., 2019), VSR (en) (Liu et al., 2023a), VisualDialog (en) (Das et al., 2017) |
| Science | AI2D (en) (Kembhavi et al., 2016), ScienceQA (en) (Lu et al., 2022a), TQA (en) (Kembhavi et al., 2017) |
| Chart | ChartQA (en) (Masry et al., 2022), MMC-Inst (en) (Liu et al., 2023c), DVQA (en) (Kafle et al., 2018), PlotQA (en) (Methani et al., 2020), LRV-Instruction (en) (Liu et al., 2023b) |
| Mathematics | GeoQA+ (en) (Cao & Xiao, 2022), TabMWP (en) (Lu et al., 2022b), MathQA (en) (Yu et al., 2023a), CLEVR-Math/Super (en) (Lindström & Abraham, 2022; Li et al., 2023c), Geometry3K (en) (Lu et al., 2021) |
| Knowledge | KVQA (en) (Shah et al., 2019), A-OKVQA (en) (Schwenk et al., 2022), ViQuAE (en) (Lerner et al., 2022), Wikipedia (en&zh) (He et al., 2023) |
| OCR | OCRVQA (en) (Mishra et al., 2019), InfoVQA (en) (Mathew et al., 2022), TextVQA (en) (Singh et al., 2019), ArT (en&zh) (Chng et al., 2019), COCO-Text (en) (Veit et al., 2016), CTW (zh) (Yuan et al., 2019), LSVT (zh) (Sun et al., 2019), RCTW-17 (zh) (Shi et al., 2017), ReCTs (zh) (Zhang et al., 2019), SynthDoG (en&zh) (Kim et al., 2022), ST-VQA (en) (Biten et al., 2019) |
| Document | DocVQA (en) (Clark & Gardner, 2018), Common Crawl PDF (en&zh) |
| Grounding | RefCOCO/+/g (en) (Yu et al., 2016; Mao et al., 2016), Visual Genome (en) (Krishna et al., 2017) |
| Conversation | LLaVA-150K (en&zh) (Liu et al., 2023e), LVIS-Instruct4V (en) (Wang et al., 2023), ALLaVA (en&zh) (Chen et al., 2024a), Laion-GPT4V (en) (LAION, 2023), ShareGPT (en&zh) (Zheng et al., 2024), SVIT (en&zh) (Zhao et al., 2023) |
| Text-only | OpenHermes2.5 (en) (Teknium, 2023), Alpaca-GPT4 (en) (Taori et al., 2023), COIG-CQIA (zh) (Bai et al., 2024), ShareGPT (en&zh) (Zheng et al., 2024) |
| Video | EgoTaskQA (en) (Jia et al., 2022), Mementos (en) (Wang et al., 2024c), STAR (en) (Wu et al., 2021), NTU RGB+D (en) (Shahroudy et al., 2016), VideoChat2IT (en&zh) (Li et al., 2023b), LSMDC-QA (en) (Rohrbach et al., 2017), ShareGPT-4o (en&zh) (Chen et al., 2024c) |
| Handwritten | SROIE (en) (Huang et al., 2019), FUNSD (en) (Guillaume Jaume, 2019), POIE (en) (Kuang et al., 2023) |

The datasets used in the instruction fine-tuning stage are listed in Tab. 7.

## A.2 MORE TRAINING DETAILS

Table 8: **Hyper-parameters used in the pre-training and instruction tuning of Mono-InternVL.**

| Configuration | Concept Learning (S1.1) | Semantic Learning (S1.2) | Alignment Learning (S1.3) | Instruction Tuning (S2) |
|---|---|---|---|---|
| Maximum numer of patches | $1,280$ | $1,792$ | $3,328$ | $6,400$ |
| LLM sequence length | $1,425$ | $1,925$ | $4,096$ | $8,192$ |
| Use thumbnail | | ✓ | | |
| Optimizer | | AdamW | | |
| Optimizer hyperparameters | | $\beta_1 = 0.9, \beta_2 = 0.999, eps = 1e^{-8}$ | | |
| Peak learning rate | $1e^{-4}$ | $1e^{-4}$ | $5e^{-5}$ | $2e^{-5}$ |
| Learning rate schedule | constant with warm-up | constant with warm-up | cosine decay | cosine decay |
| Drop path rate | | 0.1 | | |
| Weight decay | | 0.01 | | |
| Training steps | 450k | 126k | 70k | 14k |
| Warm-up steps | 100 | 100 | 100 | 420 |
| Global batch size | $2,048$ | $2,048$ | $2,048$ | $1,024$ |
| Gradient accumulation | 1 | 1 | 1 | 4 |
| Numerical precision | | bfloat16 | | |

Hyper-parameters used in the training stages are listed in Tab. 8.

### A.3 MORE ABLATION STUDIES

In Fig. 6, we provide the full results of Fig. 4 with all the downstream tasks.

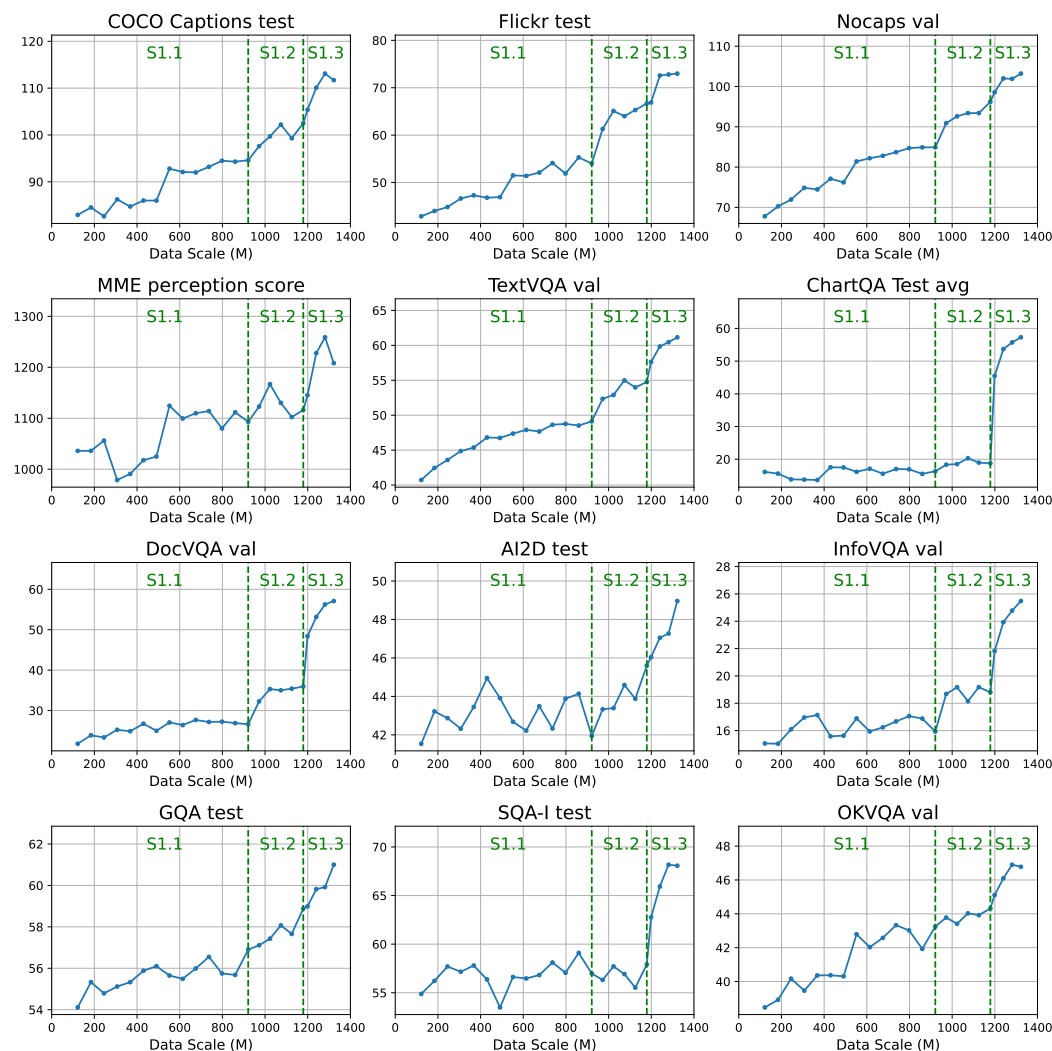

Figure 6: **Complete results of downstream performance breakdown with the increase of pre-training data size.**

In Tab. 9, we examine the effects of freezing and unfreezing attention layers in alignment learning. We observe that unfreezing attention results in consistent improvements across all metrics, suggesting that it is crucial to optimize the multi-head attentions in this sub-stage for better vision-language alignment.

Table 9: **Results of freezing and unfreezing attention in alignment learning.** "T-Param" refers to trainable parameters. All models are pre-trained on 20 millions of data in alignment learning and fine-tuned on instruction data from LLaVA-665k (Liu et al., 2023d).

| Methods | #T-Param | MME-P | DocVQA | InfoVQA | SQA-I | GQA | ChartQA | AI2D |
|---|---|---|---|---|---|---|---|---|
| Freeze attention | 1.8B | 1136 | 39.5 | 19.7 | 56.5 | 59.1 | 27.2 | 44.1 |
| Unfreeze attention | 3.0B | 1153 | 49.3 | 22.7 | 61.8 | 59.9 | 49.5 | 46.4 |

### A.4 VISUALIZATIONS

**Visual grounding**

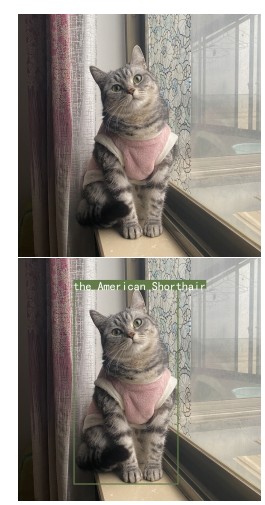

**Prompt:** Detect the <ref>the American Shorthair</ref> in the image with its bounding box.

**Response:** <ref> the American Shorthair </ref> <box> [[229, 125, 660, 932]] </box>

**Prompt:** Please describe the detected object.

**Response:** The object in the image is a cat. The cat has gray fur with black stripes and is wearing a pink sweater. It appears to be sitting on a windowsill, looking up attentively towards the camera.

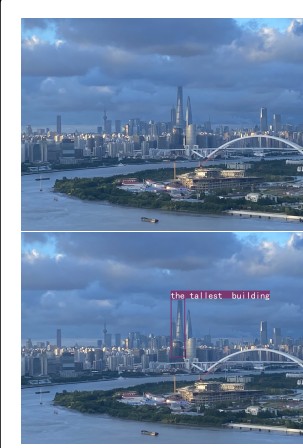

**Prompt:** Detect the <ref>the tallest building</ref> in the image with its bounding box.

**Response:** <ref> the tallest building </ref> <box> [[529, 313, 580, 591]] </box>

**Prompt:** Please infer where this place is.

**Response:** This image depicts a stunning view of the skyline of Shanghai, China. The prominent features in the skyline include several high-rise buildings with unique structures such as the Oriental Pearl Tower and the Shanghai World Financial Center. There is also a distinctive arch bridge spanning across a river, which appears to be part of Shanghai's famous Huangpu River waterfront area known for its historic sites and modern development.

**VQA**

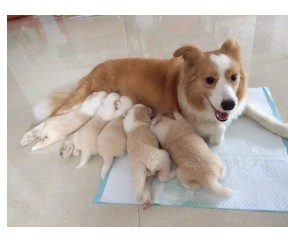

**Prompt:** Describe the image.

**Response:** The image features a large adult dog lying on its side, likely the mother of several puppies. The adult dog has brown and white fur with distinct markings on its face. She is surrounded by several small puppies, all of which are nursing from her body or resting on a light blue towel that covers part of the floor. The setting appears to be indoors, possibly in a home environment given the tiled flooring and the soft, comfortable-looking towel. The atmosphere seems calm and nurturing as the dogs interact closely together.

**Prompt:** Please tell me the exact number of golden retrievers in the picture.

**Response:** There are a total of six golden retrievers in the picture.

**Image captioning and OCR**

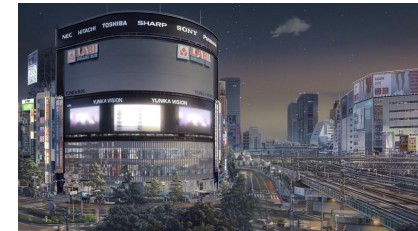

**Prompt:** Describe this image.

**Response:** This image depicts a bustling urban scene at night in an animation style, with a large electronic billboard prominently displaying the words "YUNIKA VISION" in bright white letters against a black background. The billboard is surrounded by several other tall buildings, including those of well-known companies such as HITACHI, TOSHIBA, SHARP, SONY, PANASONIC, and NEC.

In the foreground, there are multiple train tracks that intersect near what appears to be an elevated railway or subway system. Several trains are visible, indicating active transportation amidst the busy cityscape.

The skyline is dotted with various other commercial and residential structures, contributing to the sense of a thriving metropolis. Trees can also be seen around the area, providing some greenery amidst the urban environment. Overall, the image captures the dynamic energy of a modern city at night, with its blend of technology, commerce, and human activity.

**Chinese OCR**

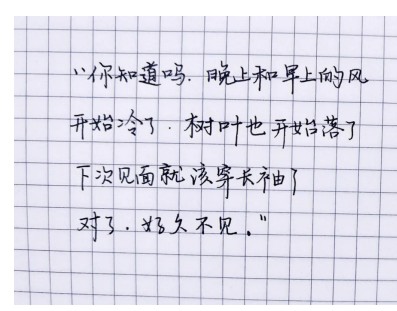

**Prompt:** 识别图中文字。

**Response:** 图中文字如下：
"你知道吗。晚上和早上的风 开始冷了，树叶也开始落了 下次见面就该穿长袖了 对了。好久不见。"

**Code generation**

$$\mathcal{L}_{self\text{-}dist}(G_{AB}, p_A) = \mathbb{E}_{\mathbf{x} \sim p_A} |\frac{1}{\sigma_A}(\|L(\mathbf{x}) - R(\mathbf{x})\|_1 - \mu_A)$$
$$- \frac{1}{\sigma_B}(\|G_{AB}(L(\mathbf{x})) - G_{AB}(R(\mathbf{x}))\|_1 - \mu_B)|$$

**Prompt:** Convert the formula in the figure to latex format.

**Response:** Certainly! Here is the formula from the figure in LaTeX format:

```
\begin{equation}
\begin{aligned}
    \mathcal{L}_{self-dist}(G_{AB}, p_A) = \mathbb{E}_{\textbf{x}\sim p_A}|
    \frac{1}{\sigma_A}(|| L(\textbf{x}) - R(\textbf{x}) ||_1 - \mu_A)\\
    - \frac{1}{\sigma_B}(||G_{AB}(L(\textbf{x})) - G_{AB}(R(\textbf{x}))||_1 - \mu_B)|
\end{aligned}
\end{equation}
```

**Document understanding**

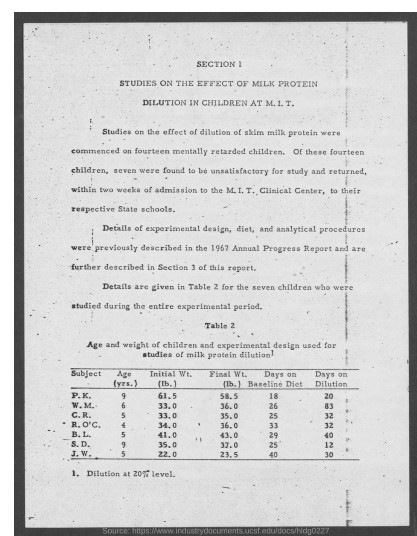

**Prompt:** What is the age of subject B.L.?

**Response:** B.L. is 5 years old at the time of study on the effect of milk protein dilution.

**Prompt:** Whose initial weight is the largest? What is it?

**Response:** The largest initial weight in the table is for P.K. with 61.5 pounds (lb.).

**Prompt:** How many children participated in the experiment in total?

**Response:** There are a total of 14 children who participated in the experiment.

**Math**

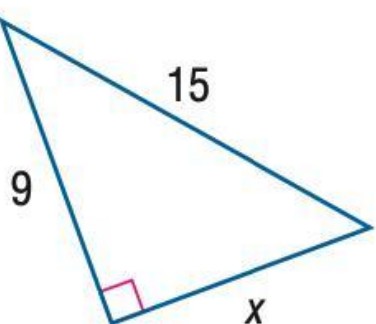

**Prompt:** The figure shows a right-angled triangle. What is the value of $x$?

**Response:** To find the value of $x$, we can use the Pythagorean theorem, which states: $a^2 + b^2 = c^2$ where c is the hypotenuse (the side opposite the right angle), and a and b are the other two sides.
In this triangle:
The length of the hypotenuse ($c$) is 15.
One leg ($a$) is 9.
The other leg ($b$) is $x$.
Using the Pythagorean theorem:
$9^2 + x^2 = 15^2$
$81 + x^2 = 225$
Now, solve for $x$:
$x^2 = 225 - 81$
$x^2 = 144$
$x = \sqrt{144}$
$x = 12$
So, the value of $x$ is 12.

