# OpenReview forum: "Mono-InternVL: Pushing the Boundaries of Monolithic Multimodal Large Language Models with Endogenous Visual Pre-training"
_ICLR.cc/2025/Conference — ICLR 2025 Conference Withdrawn Submission_

### Official Review · Reviewer_y1bf · 2024-10-30

**Soundness:** 3
**Presentation:** 3
**Contribution:** 2
**Rating:** 5
**Confidence:** 5

**Summary:**

This paper presents Mono-InternVL, a strong monolithic MLLM with multimodal MoE that directly embeds images and texts into patches and tokens without widely-used visual encoders in MLLMs.
Large-scale multi-stage multimodal pretraining and SFT helps Mono-InternVL achieve strong performance on various benchmarks, compared with both monolithic and
modular MLLMs.
Stands on the shoulder of previous works, Mono-InternVL provides a feasible solution to extend
pre-trained LLMs to monolithic MLLMs.

**Strengths:**

1. This paper conducts large-scale multimodal pretraining and SFT to train a monolithic MLLM with multimodal MoE. The well-designed model architecture and multi-stage training strategy help Mono-InternVL achieve strong performance on various benchmarks, compared with both monolithic and modular MLLMs.
2. This paper provides the scaling curve of multi-stage pre-training data and the performance of Mono-InternVL on downstream tasks, which is helpful for understanding the impact of data scale on different training stages.
3. This paper provides a visualization of attention maps to help understand the attention mechanism of such a monolithic MLLM between text and image modalities.

**Weaknesses:**

1. The main contribution of this paper is to extend pre-trained LLMs to monolithic MLLMs via multimodal MoE and multi-stage pre-training. However, the novelty of this work is limited, as it mainly relies on previous works, especially VLMo, which proposed multimodal MoE and stagewise pre-training in the VLMs era. With full respect to the author's contribution as wrote in the strengths, but Mono-InternVL is very much a VLMo that combines the experience of data and training engineering tricks widely used in existing MLLMs.
2. This paper lacks fair comparison and in-depth analysis on modular and monolithic MLLMs. As a counterfact to Mono-InternVL, the authors should provide a modular version (Modular-InternVL) to fairly compare the performance (Table 2,3,4). Take Table 2 as an example, compared with other modular MLLMs, too many factors that have a significant impact on performance make such comparisons unfair and unconvincing (LLM backbone, training data, evaluation setting like image resolution).
3. This paper seems to avoid comparing with strong modular and monolithic MLLMs, such as InternVL2, Qwen2-VL, Phi-3-Vision, and Reka (a very strong monolithic MLLM). These comparisons are meaningful and important. It is very strange that the authors ignore these comparisons, especially when InternVL2 shares the same LLM backbone with Mono-InternVL, and the LLM backbone has a significant impact on performance.

**Questions:**

1. The Avg score used in Table 2 is very strange. Most of the models in the table have never seen Chinese multimodal data, or even Chinese text data. In contrast, Mono-InernVL uses Chinese multimodal data in both pre-training and fine-tuning stages. In this case, introducing a Chinese multimodal evaluation dataset into the calculation of the Avg score is a very unfair and strange behavior.
2. Mono-InternVL's training data includes InternVL-1.5's training data, and the high-resolution image processing strategy is also inherited from InternVL-1.5. However, there is a significant gap in performance on high-resolution benchmarks (DocVQA, InfoVQA in Table 3), even InternVL-1.5 uses a weaker LLM backbone. Could you provide more dicussion and analysis on this issue?
3. Considering authors claim about catastrophic forgetting, it is very interesting to compare the text-only performance of the LLM backbone and Mono-InternVL, to verify the claim.
4. In Table 6, the authors compare the decoding speed of Mono-InternVL with InternVL-1.5 via TTFT and TPS. However, the introduction of MMoE may slow down the decoding speed LLM part in the multimodal input, especially image-text interleaved settings. When the model scale and output token number are small, the speed advantage of Mono-InternVL without visual encoder can be highlighted. However, when the model scale and output token number are large, the speed advantage of Mono-InternVL may be weakened and the slower decoding speed of MMoE may become an important issue that deserves attention. Could you provide more discussion and analysis on this issue? For example, analyzing the decoding speed (i.e., for each sample, TTFT + Decoding Time = Processing Time of each sample) of Mono-InternVL and InternVL-1.5 in both simple image-text setting and complex image-text interleaved setting when parallel generating answers.
5. In Sec. 3.1, what's the meaning of "add a thumbnail to provide global visual information" when transforming the image to image patches?

---

### Official Review · Reviewer_KTLo · 2024-11-03

**Soundness:** 3
**Presentation:** 3
**Contribution:** 3
**Rating:** 5
**Confidence:** 5

**Summary:**

This paper present Mono-InternVL, a novel monolithic MLLM that seamlessly integrates a set of visual experts via a multimodal mixture-of-experts structure. Besides, they leverage a innovative pre-training strategy to maximize the visual capability.  Extensive experiments on 16 benchmarks validate its superior performance.

**Strengths:**

I have a very positive and encouraging attitude toward the construction of native MLLM models. I believe this work is a very meaningful exploration of Monolithic MLLM. The model structure is relatively simple, yet the performance achieved is quite significant. The writing of the paper is also clear and easy to follow.

**Weaknesses:**

* **Confusion in Structural Design:** In L314-315 of the manuscript, it is mentioned that unfreezing the parameters of the MHA component during the **Alignment Learning** stage helps achieve better vision-language alignment. However, this improvement does not seem intuitive. Furthermore, will unfreezing the MHA while freezing the T-Ext lead to potential losses in language capability?

* **Scalability of the Method:**
    * Compared to mainstream paradigms based on ViTs, the monolithic architecture requires a substantial amount of image-text data to facilitate endogenous visual pre-training. My concern is that, unlike ViTs which can adapt to various LLMs through lightweight image-text alignment (*e.g.*, LLaVA 558K), the visual expert trained within Mono-InternVL is specific to a particular LLM. Specifically, **The semantic space and model structure of the visual expert are both specialized for InternLM**, making it difficult to reuse them with other LLMs (e.g., Qwen, LLaMA3). It significantly limits the scalability of the approach, especially considering the high training costs involved.
    * Additionally, performance validation was only conducted on a 1.8B parameter scale. While the performance is notable compared to other ViT-based models of similar size, scaling up to 7B or even 72B while continuing to utilize a visual expert reveals a clear disadvantage in training costs compared to ViT-based models.

* **Fairness in Performance Comparison:** Another concern arises from Table 6. In the comparison of computational overhead, is it truly necessary to only compare with InternVL-1.5-2B? For instance, as the authors demonstrate in Table 3, several ViT-based models perform comparably to Mono-InternVL under similar activation parameters (e.g., MiniCPM-V-2). To my knowledge, these models utilize only SigLiP or CLIP-L as their image encoder, resulting in a minimal number of image tokens. In contrast, Mono-InternVL employs a significantly larger number of patches. Therefore, I believe a comprehensive comparison of performance and training costs between these models and the proposed Mono-InternVL is necessary to accurately reflect the advantages of the monolithic model.

I hope the author can provide a more detailed explanation and proof of the proposed method. I will consider changing my perspective if the author's response is valuable.

**Questions:**

Please refer to Weaknesses.

---

### Official Review · Reviewer_1gyt · 2024-11-03

**Soundness:** 3
**Presentation:** 3
**Contribution:** 2
**Rating:** 5
**Confidence:** 5

**Summary:**

The paper enhances monolithic MLLMs by employing delta tuning, which prevents the model from suffering catastrophic forgetting. Building on this, it integrates a set of visual experts using a MoE structure and introduces a novel pre-training strategy EViP for progressive learning of visual knowledge. Experimental results in the paper show that the proposed framework achieves superior performance across various benchmarks.

**Strengths:**

- The paper tackles the significant challenge of catastrophic forgetting in MLLM foundation model training. By utilizing delta tuning approach within monolithic MLLMs, it mitigates this problem, ensuring the retention of core language capabilities while learning new visual parameters.
- Given the large-scale nature and critical importance of data quality in pre-training and instruction tuning, the authors' introduction of a progressive learning strategy is commendable.
-  The paper is supported by extensive experimental validation, demonstrating the framework’s performance across 16 benchmarks.

**Weaknesses:**

- A key weakness of the paper lies in the lack of clear motivation for adopting monolithic MLLMs over modular MLLMs. Understanding the advantages and rationale behind choosing monolithic structures is crucial for assessing the significance and justification of the work. The authors need to provide a detailed explanation of why this approach is preferable and what specific benefits it offers compared to modular MLLMs.
- The method proposed in the paper lacks sufficient novelty. To address catastrophic forgetting, the paper introduces independent visual parameters, an approach that is already widely used in modular MLLMs such as QwenVL and InternVL-1.5. The authors need to clearly explain how their approach differs from these existing methods. Simply transferring these techniques from modular to monolithic MLLMs does not constitute a significant innovation.
- The paper's approach to processing visual input in monolithic MLLMs employs PatchEmbed and MLP, while modular MLLMs typically use a visual encoder and MLP. PatchEmbed, as a learnable module, effectively encodes visual input into tokens that can be fed to the LLM, similar to the role of a visual encoder. This raises the question of how the proposed method truly differentiates itself from modular approaches, as both share the fundamental idea of transforming visual input into LLM-readable tokens. The authors need to clarify how this design embodies the monolithic concept and what the essential differences are compared to modular MLLM structures.
 - The paper introduces a progressive learning approach that divides the pre-training process into three distinct stages. To substantiate the effectiveness of this method, the authors should provide ablation studies demonstrating its impact. Additionally, it would be beneficial to explain why the pre-training data is segmented into these three categories, potentially from the perspective of data distribution. Moreover, it raises the question of whether instruction-tuning data would also benefit from a similar progressive approach.
- The benchmarks used for comparison in the paper are incomplete, as they do not include prominent MLLMs such as LLaVA-1.5 and Qwen2-VL. The absence of these mainstream models weakens the claim of superiority for the proposed method.
 - The authors need to compare MLLMs of different sizes to demonstrate the scalability of their approach. Currently, the models evaluated in the paper are all under 2B parameters. To validate the proposed method's applicability and performance at scale, it is essential to include experiments with larger models, such as 7B and 13B.
- The paper lacks detailed implementation specifics for FFN_v and FFN_t. It states that FFN_v is initialized from FFN_t to leverage pre-trained knowledge. The author should provide a comprehensive explanation of the implementation process. This detail is crucial for understanding the decoupling and interaction between visual and textual knowledge.

**Questions:**

Refer to the weakness.

---

### Official Review · Reviewer_JGVY · 2024-11-04

**Soundness:** 3
**Presentation:** 3
**Contribution:** 2
**Rating:** 6
**Confidence:** 3

**Summary:**

This paper introduces a novel monolithic Multimodal Large Language Model (MLLM) architecture, which integrates visual parameters directly into a pretrained Language Model through a multimodal mixture-of-experts (MoE) framework, eliminating the need for additional visual feature extractors and projectors. Additionally, a delta tuning pretraining strategy, named EViP, is proposed to address the issue of catastrophic forgetting. Experiments conducted on 16 benchmarks demonstrate the superiority of both the proposed model and the pretraining strategy.

**Strengths:**

1. The monolithic architecture presented in this paper is highly innovative and relatively easy to reproduce. The MMoE framework not only allows the visual learning components to benefit from the pretrained language sections but also maintains linguistic capabilities. This dual advantage significantly accelerates the monolithic pretraining process.
2. The EViP pretraining strategy is well-motivated, enabling smoother pretraining through progressive learning, which addresses catastrophic forgetting effectively.
3. The paper includes extensive experiments and ablation studies, providing compelling evidence for the effectiveness of the proposed method.

**Weaknesses:**

1. While the paper introduces several key contributions, such as the multimodal MoE architecture and the progressive training strategy, it does not clearly delineate the performance gains attributable to each component.
2. The concept of a multimodal MoE structure has been explored in prior works such as MoE-LLaVA, MARS, and Uni-MoE. However, the paper lacks a detailed discussion on how this model differs from these existing approaches.
3. In Section 3.3, the authors mention the inclusion of additional data sets for video understanding and handwritten text recognition. However, the paper does not clarify the impact of these data sources on the model's performance.

[1] MoE-LLaVA: Mixture of Experts for Large Vision-Language Models

[2] MARS: Mixture of Auto-Regressive Models for Fine-grained Text-to-image Synthesis

[3] Uni-MoE: Scaling Unified Multimodal LLMs with Mixture of Experts

**Questions:**

See Weaknesses

---

### Note · Authors · 2024-11-13

I have read and agree with the venue's withdrawal policy on behalf of myself and my co-authors.